# Impact of Arsenic Stress on the Antioxidant System and Photosystem of *Arthrospira platensis*

**DOI:** 10.3390/biology13121049

**Published:** 2024-12-15

**Authors:** Jiawei Liu, Jie Du, Di Wu, Xiang Ji, Xiujuan Zhao

**Affiliations:** 1College of Life Sciences, Inner Mongolia Agricultural University, Hohhot 010010, China; 13212202636@163.com; 2Bayannur Center for Disease Control and Prevention, Bayannaoer 015000, China; 15147266931@163.com; 3Department of Chemical and Environment Engineering, Hetao College, Bayannaoer 015000, China; wudhtdx@163.com

**Keywords:** *Arthrospira platensis*, arsenic stress, antioxidant system, PSII, transcriptomics

## Abstract

Arsenic contamination in groundwater poses a severe threat to public health in many countries and regions worldwide. Therefore, it is crucial to reduce arsenic levels in water at the source. Most research has focused on the role of aquatic organisms in arsenic detoxification. In this study, we investigated the arsenic resistance mechanisms of *Arthrospira platensis*. By measuring growth, cellular activity, photosynthetic systems, and antioxidant system indicators, we elucidated the arsenic detoxification mechanisms of *Arthrospira platensis* cells under both control and arsenic- treated conditions. Additionally, we conducted transcriptome sequencing and subsequent qPCR experiments to validate our findings. The results demonstrate that *Arthrospira platensis* exhibits high tolerance to arsenic and mitigates arsenic toxicity through valence conversion, offering a potential solution to the problem of arsenic contamination in groundwater.

## 1. Introduction

Arsenic (As) poses a considerable threat to human health and has been classified as a Group 1 carcinogen by the International Agency for Research on Cancer. Previous studies have shown that long-term consumption of water with excessive arsenic levels can cause a range of different damages to the nervous system, skin, and liver [1] Moreover, acute ingestion of arsenic-contaminated water can lead to arsenic poisoning, which poses a life-threatening risk [1]. Inorganic arsenic exists in nature in forms such as arsenic, elemental arsenic, arsenites, and arsenates. In organisms capable of bioaccumulation, arsenic typically exists in organic forms, such as methylated arsenic, arsenosugars, and arsenolipids [2]. Arsenic is non-degradable in the environment and is absorbed by aquatic organisms through ingestion. It is bioaccumulated in higher trophic-level organisms via the food chain and food web [3]. More than 70 countries in the world have an excessive amount of arsenic in their water, and the Inner Mongolia Autonomous Region of China is one of the areas most seriously affected by drinking-water arsenic exceeding the limit [4].

In recent years, there has been a growing interest in studying plant resistance to arsenic, particularly in edible crops, which are pivotal in the arsenic contamination of the food chain [5]. Some studies have proposed that genetic engineering techniques can be employed to enhance plants’ tolerance to arsenic by improving their ability to bind this metalloid [6]. Research has demonstrated that by modulating genes associated with glutathione and phytochelatins, it is feasible to significantly augment the arsenic-binding capacity of plants [7].

Previous studies have demonstrated that *A. platensis* can coexist with arsenic, which exhibits a high tolerance to arsenic stress [8,9]. When *A. platensis* is stressed by arsenic, it produces a large amount of reactive oxygen species (ROS) with cytotoxic effects [10]. There are two evolved antioxidant systems in *A. platensis*: the antioxidant enzyme system and the non-enzymatic antioxidant system (including non-enzymatic antioxidants such as glutathione and ascorbic acid), both of which are pathways for the removal of ROS [11]. Previous studies on the *A. platensis* stress response to heavy metals have indicated that the detoxification mechanisms of microalgae include cell surface binding, redox reactions, methylation, and other processes. Algae can accumulate arsenic compounds, and arsenic can bind to the cell surface or form complexes with phytochelatins inside the cell, leading to precipitation and subsequent excretion [12]. In addition, microalgae oxidize highly toxic trivalent arsenic to low-toxicity pentavalent arsenic via arsenite oxidase [13]. Conversely, microalgae reduce pentavalent arsenic to trivalent arsenic by methylation or complexation when the As^5+^ is in excess [14]. Trimethylarsenic (TMA) is a volatile substance with low toxicity at low concentrations. It is considered a detoxification strategy for microalgae because its volatilization reduces arsenic levels in cells [15]. Moreover, there are arsenic efflux mechanisms in microalgae and other arsenic-detoxifying microorganisms, typically relying on arsenic efflux pump proteins such as ArsB, Acr3, ArsJ, ArsP, MSF1, and ArsK [16]. These organisms possess multiple arsenic transporters (regulated by different operons) corresponding to the transport of different arsenic compounds [16].

However, previous studies have only focused on physiological and biochemical indicators and have not delved into the specific mechanisms of arsenic detoxification. To investigate the resistance mechanisms of *A. platensis* to arsenic, we designed an experiment comparing a control group and an arsenic-stressed group through physicochemical indicators and transcriptomic analysis. We elucidated the arsenic detoxification mechanisms in *A. platensis* cells by measuring the growth, cellular active substances, photosynthetic system, and antioxidant system indicators between the control conditions and arsenic treatment. Furthermore, we performed transcriptome sequencing and subsequent q-PCR experiments to validate the results. Our findings ultimately confirmed the arsenic-resistance mechanisms and effectiveness of *A. platensis*, which demonstrated high tolerance and can mitigate arsenic toxicity through valence conversion, offering a potential solution for addressing arsenic contamination in groundwater.

## 2. Materials and Methods

### 2.1. Strain and Cultivate Condition

*Arthrospira platensis* (XJ007) was obtained from the Inner Mongolia Key Laboratory of Biomass-energy Conversion. The cells were cultured in 250 mL Erlenmeyer flasks containing improved Zarrouk culture medium (Appendix A), with a diurnal cycle of 12/12 h light/dark at 28 °C light intensity 3800 lux [8]. For stress treatments, sodium arsenite standard solution at initial concentrations ranging from 0 to 150 mg/L was added to the medium when the cells reached OD_560_ = 0.2 ± 0.01 (Day 1). All experiments were performed in triplicate.

### 2.2. Determination of A. platensis Growth Curve

The UV spectrophotometer was used to measure OD_560_ every two days for 20 days (0 days, 2 days, 4 days, 6 days …20 days) and plot the growth curve with days as the *X*-axis and the OD_560_ value as the *Y*-axis.

### 2.3. Determination of Active Substances in A. platensis Cells

The determination of phycocyanin was conducted according to Han [8]. Specifically, algae cultures in the logarithmic growth phase (Day 8) were adjusted to a uniform OD, and 15 mL aliquots were taken. These aliquots were then centrifuged (12,000 r/min, 4 °C, 15 min) to collect the algal pellet in 50 mL centrifuge tubes. Subsequently, 15 mL of PBS buffer (0.1 mol/L, pH 7.8) was added to the pellet, which was then vortexed. The mixture was subjected to repeated freeze–thaw cycles at −80 °C and 25 °C, with fixed durations (30 min at −80 °C; 15 min at 25 °C), repeated 6 times. After the final cycle, the sample was centrifuged again (12,000 r/min, 4 °C, 10 min), and the supernatant was collected for measurement of OD615 and OD652 to calculate the concentration of phycocyanin [17].
PC (mg/mL) = 0.187 × OD_615_ − 0.089 × OD_652_

Extraction of chlorophyll a and carotenoids from *A. platensis* involved using the organic solvent immersion method [8]. The algal pellets collected were the same as 2.3 in 50 mL centrifuge tubes. Next, 20 mL of 96% ethanol was added, and the mixture was vigorously shaken. The tubes were then wrapped in foil and placed in a 4 °C refrigerator for 24 h, with intermittent shaking every 6 h. The samples were centrifuged again (8000 r/min, 4 °C, 10 min), and the supernatants’ OD665, OD649, and OD470 were measured to calculate the concentrations of chlorophyll a and carotenoids using the formula provided in reference [18].
Chlorophyll a: Ca (mg/L) = 9.78 × 4OD_662_ − 0.990 × OD_644_

Chlorophyll b: Cb (mg/L) = 21.426 × OD_644_ − 4.650 × OD_663_

Carotenoids: Cc (mg/L) = 4.695 × OD_440_ − 0.268 × (Ca + Cb) 

### 2.4. Determination of the Photosynthetic System

The 5 mL logarithmic growth phase was adjusted to uniform OD value in 0.1 ± 0.01 by the medium. Next, 3 mL of the algae suspension was placed in a glass test bottle and subjected to dark treatment for 30 min in a dark environment, followed by fluorescence parameter measurements under no-light conditions. The plant photosynthetic efficiency meter (Handy PEA, Norfolk County Britain Hansatech) was used to measure the fluorescence parameters and analyze the damage to photosystem II.

### 2.5. Determination of the Antioxidant System in A. platensis Under As^3+^ Stress

For each group, 45 mL of the solution with the same OD value was centrifuged at 12,000 r/min for 15 min, and the algal paste was collected in a mortar. Then, it was ground on ice after adding 2–4 g of silica. The mixture was washed by 0.05 mol/L PBS and ground a second time. The solution was centrifuged at 4 °C, 8000 r/min, for 10 min and stored in the suspension. In 4 °C. The suspension was used to measure the activities of superoxide dismutase (SOD) and catalase (CAT) as well as the intracellular malondialdehyde (MDA) content using a Nanjing Jiancheng reagent kit (A001-1-2, A003-1-2, A007-1-1,Nanjing, China).

### 2.6. Extraction of RNA, Synthesis of cDNA, and Transcriptome Sequencing

The total RNA of the 70 mg/L arsenic treatment group and the control group was extracted using TRIzol reagent (Thermo Fisher Waltham, MA, USA). Ribosomal RNA (rRNA) depletion instead of poly(A) purification was performed by the RiboCop rRNA Depletion Kit for Mixed Bacterial Samples (Lexogen, NH, USA), and then, all mRNAs were broken into short (200 nt) fragments by adding fragmentation buffer first. Secondly, double-stranded cDNA was synthesized with random hexamer primers (Illumina). When the second-strand cDNA was synthesized, dUTP was incorporated in place of dTTP. Then, the synthesized cDNA was subjected to end-repair, phosphorylation, and ‘A’ base addition according to Illumina’s library construction protocol. The RNA-seq transcriptome library was prepared following Illumina^®^ Stranded mRNA Prep, Ligation (San Diego, CA, USA) using total RNA. The paired-end RNA-seq library was sequenced with the Illumina Novaseq 6000 (or other new sequenator) (Illumina Inc., San Diego, CA, USA). The original images were processed for sequences, base-calling, and quality value calculations. Clean reads were generated by removing low-quality sequences reads with more than 10% of N bases (unknown bases) and reads containing adaptor sequences.

### 2.7. qPCR Validation

Based on the transcriptome sequencing results, a primer designed for the target gene was conducted using the Primer-BLAST tool provided on the NCBI database platform. The primer sequences are listed in Appendix A. The SYBR Green ProTaq HS Premix qPCR Kit (Accutar Biotechnology, Shanghai, China) was used for amplification, with the following program: 95 °C for 30 s, followed by 40 cycles of 95 °Cfor 5 s and 60 °C for 34 s. After the cycles, the program included 95 °C for 15 s, 65 °C for 0.05 s, and 95 °C for 0.5 s. The RT-qPCR procedure employed the two-step method, and the differential expression levels were calculated using the 2^−ΔΔCT^ algorithm.

### 2.8. Data Processing and Analysis

Data processing was performed by IBM SPSS 26, and the LSD was employed for differential analysis (*p* < 0.05 indicating significant differences). GraphPad Prism 9 was used for plotting, with each sample undergoing three biological replicates. Error bars represent standard deviation (SD).

## 3. Results

### 3.1. The Growth and Biological Activity of A. platensis

Figure 1a illustrates the growth of *A. platensis* cells under As^3+^. We found that there was a biphasic effect in *A. platensis* cells. Briefly, the growth rate of the cells under 20 mg/L As^3+^ was faster than the control group. In contrast, at high As^3+^ concentration, the growth of the cells was significantly inhibited. Figure 1b demonstrates that the phycocyanin contents decreased significantly with increasing As^3+^ concentration across the experimental groups (*p* < 0.05). The results demonstrate that the contents of carotenoid (Figure 1c) and chlorophyll (Figure 1d) in *A. platensis* were decreased under As^3+^ stress. We found the contents of carotenoid and chlorophyll were significantly decreased when the As^3+^ concentration exceeded 20 mg/L compared to the control group (*p* < 0.05). However, within the range of 20 mg/L to 90 mg/L As^3+^, there were no significant differences in the contents of carotenoid and chlorophyll a. A highly significant decrease in both carotenoid and chlorophyll was observed in the groups treated with 150 mg/L As^3+^.

### 3.2. The Photosynthetic System of A. platensis

Figure 2a shows the standard OJIP curve shapes for all treatments. Compared to the control group, the I and P points of the curves under As^3+^ stress show a decrease, which indicates that arsenic caused certain damage to the photosynthetic system of *A. platensis*. The damage was intensified with increasing As^3+^ concentration (Figure 2a). Chlorophyll fluorescence transients (Fv/Fm, ETo/RC, TRo/RC, ABS/CS, ABS/RC, Sm, Vj, Mo, Wk, and Plabs) were deduced through OJIP curve fittings. The results in Figure 2b show that the parameters ABS/RC, Vj, and Mo increased with increasing arsenic concentration. Meanwhile, other parameters decreased with increasing arsenic concentrations, such as Fv/Fm, PIabs, Wk, Sm, ABS/CS, and ETo/RC. These results suggest that the electron transfer in the chloroplasts of *A. platensis* cells was damaged by arsenic stress. Moreover, the number of active reaction centers and the energy transferred by electrons was reduced.

### 3.3. The Antioxidant System of A. platensis

The activities of SOD and CAT were significantly higher in 20 mg/L to 60 mg/L of As^3+^ concentration than those in the control group (Figure 3a, b) (*p* < 0.05). However, as the As^3+^ concentration increased from 100 mg/L to 150 mg/L, the activities of both enzymes were significantly lower than those of the control group (*p* < 0.05). The results of the MDA contents indicated that lipid peroxidation of *A. platensis* cell membranes continued to increase with increasing As^3+^ concentration, which resulted in damage to the cell membrane and the inner mitochondrial membrane (Figure 3c).

### 3.4. Transcriptome Sequencing Analysis and Quantitative Analysis of Key Genes

#### 3.4.1. Transcriptome Sequencing Results

According to the sequencing results, the similarity between samples was analyzed by using the clustering method to calculate the distance between samples according to gene expression (Figure 4a,b). Samples within groups were well clustered in control groups and treatment groups (Figure 4a). The heatmap shows that the transcription levels of samples in a given group were similar, and there was a large difference between the groups (Figure 4b). There were a total of 177 differently expressed genes between the control groups and treatment groups, of which 90 genes were upregulated (marked in red), and 87 genes were downregulated (marked in blue) (Figure 4c).

The results of the KEGG enrichment analysis indicate that all differentially expressed genes related to metabolism were associated with porphyrin and chlorophyll metabolism as well as oxidative phosphorylation. In terms of environmental information processing, differentially expressed genes were mainly related to the bacterial secretion system, and in terms of genetic information processing, differentially expressed genes were mainly related to ribosomes (Figure 5a). Meanwhile, the upregulated differentially expressed genes associated with metabolism were primarily related to porphyrin and chlorophyll metabolism, oxidative phosphorylation, the TCA cycle, and pyruvate metabolism (Figure 5b). In contrast, the downregulated differentially expressed genes in metabolism were mainly associated with carbapenem biosynthesis, while those related to organismal systems were primarily involved in plant–pathogen interaction (Figure 5c).

GO functional enrichment revealed that oxidoreductase activity, hydrogen dehydrogenase activity, structural molecule activity, metal cluster binding, iron–sulfur cluster binding, and four iron–four sulfur cluster binding in *A. platensis* cells significantly changed (Figure 6a).

The results of the enrichment analysis showed that the activities of hydrogen donor-dependent oxidoreductases and other oxidoreductases in *A. platensis* cells significantly increased (Figure 6b) under arsenic. These results indicate that the antioxidant system of *A. platensis* was activated after being stressed by As^3+^. Meanwhile, the results of Figure 6b indicate that there were significantly upregulated genes related to [4Fe-4S] binding proteins, iron–sulfur cluster binding proteins, metal cluster linkage proteins, tetrapyrrole biosynthesis, porphyrin-containing compound biosynthesis, and metabolic processes. Iron–sulfur cluster binding proteins were vital for the chloroplast electron transport chain and contributed to 7-hydroxymethyl chlorophyll a reductase (HCAR) in the chlorophyll cycle [19]. Additionally, metal cluster linkage proteins, tetrapyrroles, and porphyrin-containing compounds were all related to metal ion binding [20]. These findings suggest that *A. platensis* binds intracellular As^3+^ following arsenic stress. The upregulation of chlorophyll metabolism, chlorophyll biosynthesis processes, light-independent chlorophyll biosynthesis, pigment biosynthesis, and metabolic processes indicates an acceleration of energy metabolism to counteract arsenic stress. Furthermore, increased synthesis of cell membranes and ribosomes implies that *A. platensis* enhances its membrane synthesis, while the upregulation of metal ion binding proteins may facilitate arsenic efflux.

Gene enrichment analysis revealed that the significantly decreased activities included transferase activity, membrane, inorganic molecular entity transmembrane transporter activity, metal ion transmembrane transporter activity, external encapsulating structure organization, cell wall organization or biogenesis, and cell wall organization in *A. platensis* under arsenic stress (Figure 6c). This indicates that the transmembrane transport system was damaged.

#### 3.4.2. Effects of As^3+^ Stress on Key Genes of the Oxidative Phosphorylation Pathway and the Phaeophytin-a Metabolism Pathway

The sequencing results of the oxidative phosphorylation pathway (Figure 7a) indicated that one NADH dehydrogenase was downregulated after As^3+^ stress, while three NADH dehydrogenases (Hox family) were upregulated. These findings suggest that the *A. platensis* enhances Hox family expression to mitigate the reactive oxygen species (ROS) generated by oxidative stress induced by arsenic. Similarly, the expression of *hoxE*, *hoxU*, and *hoxF* were upregulated under arsenic stress (Figure 7b). The *HoxEFU* complex is involved in hydrogen production and electron transport regulation in *A. platensis.*

The expression of two enzymes (*chlM* and *chlB*) involved in the chlorophyll a synthesis pathway in *A. platensis* was elevated (Figure 8a), indicating that the cells were accelerating the synthesis of chlorophyll a. We also found that the expression of *cydB* significantly decreased (Figure 8b). This suggests that there was damage in the photosynthetic system of *A. platensis* by As^3+^.

In Figure 9a, we found that the expression of *Trxs* genes in the treatment group increased compared to the control. The *MgtC protein family* plays a crucial role in Mg^2+^ transport, which is essential for the core component of chlorophyl ll. The qPCR result showed that the expression of *MgtC protein family* genes significantly decreased (Figure 9b).

## 4. Discussion

### 4.1. Response Pathways of A. platensis Photosystems to As^3+^ Stress

In this study, there was a significant decrease in the levels of phycocyanin, chlorophyll a, and carotenoids in *A. platensis* after being As^3+^ stressed (Figure 1b–d). This finding is consistent with the results of Gan et al. [21]. Previous research indicated that phycocyanin, carotenoids, and chlorophyll a are crucial light-harvesting pigments in algal cells [8]. Meanwhile, chlorophyll a and carotenoids are core components of the photosynthetic system and are essential for the process of photosynthesis. Consequently, when the photosynthetic system of *A. platensis* is compromised, the result is reduced photosynthetic efficiency under As^3+^ stress. Some research showed that the significant decrease in chlorophyll a and b contents suggests an impairment in the photosynthetic system of *A. platensis* caused by TM exposure [22].

The MgtC protein family plays a crucial role in Mg^2+^ transport, which is essential for the core component of chlorophyll. Under arsenic stress at 70 mg/L, the expression of MgtC protein family genes significantly decreased (Figure 9b), leading to reduced Mg^2+^ transport and subsequently affecting chlorophyll synthesis. This reduction in chlorophyll synthesis weakens photosynthesis and lowers the efficiency of light energy conversion and transfer. Additionally, previous studies [23] have shown that MgtC proteins can inhibit ATP generation by binding to ATP synthase. In this study, the decrease in MgtC expression may prompt *A. platensis* to increase ATP production as a response to arsenic stress.

Additionally, ABS/RC, Vj, and Mo all increased with increasing arsenic concentrations in this study (Figure 2b). This may be due to the increased concentration of As^3+^ accelerating the rate of reduction in QA while simultaneously impeding the electron transfer from QA to QB. The stress induced by As^3+^ disrupted the oxygen-evolving complex (OEC), thereby hindering electron transfer from QA to QB [24]. The findings of Wang et al. [25] similarly indicated that more QA was reoxidized via S2(QAQB) charge recombination when electron transfer from QA to QB was inhibited by As (III). As (III) may result in an increased stability of the S2QB and S2QA recombination.

This study also found that the photosynthetic efficiency was reduced by As^3+^ stress in *A. platensis* cells. Fv/Fm represented the maximum quantum yield of PSII photochemistry and quantified the maximum photochemical efficiency of open reaction centers. PIabs is a key indicator of the total number of activated photochemical reaction centers in PSII, which reflects chloroplast activity [26]. In this study, both of these indices significantly decreased under As^3+^ stress. Relevant studies have shown that short-term exposure to radionuclides led to a decrease in Fv/Fm and PIabs indices in *Anacystis nidulans* with the increase in exposure time and concentration [27], consistent with the results of this experiment. The Wk value directly reflects the condition of the oxygen-evolving complex (OEC) in PSII, which plays a crucial role in the electron transport of the photosynthetic system. It was found that under As^3+^ stress, the Wk value on the donor side of *A. platensis* PSII significantly decreased (*p* < 0.05), indicating that the donor side of *A. platensis* PSII was damaged, and the electron transport process was inhibited. Similarly, Misra et al. and Kuwabara et al. also obtained the same results [25,28]. The Sm value reflected the size of the electron acceptor pool PQ on the acceptor side of *A. platensis* PSII. A decrease in the Sm value affects the electron transport of PSII and also indicates damage to the PQ pool. When the As^3+^ concentration was 20 mg/L, the Sm value showed no significant change compared to the control group; when the As^3+^ concentration was ≥50 mg/L, the Sm value significantly decreased (*p* < 0.05). This suggests that PQ pools and electron transport were not impaired at low concentrations of As^3+^ stress, and PQ pools were damaged at high concentrations of As^3+^ stress.

### 4.2. Response Pathways of A. platensis Antioxidant System to As^3+^ Stress

C-phycocyanin is essential for enhancing antioxidant capacity in response to heavy metal stress. Liu et al. [29] investigated the physiological responses of *A. platensis* to PB^2+^, reporting a decrease in the content of C-phycocyanin as PB^2+^ concentration increased from 10 to 50 mg/L. This finding is consistent with the results obtained in this study (Figure 1b). Some studies have demonstrated that C-phycocyanin is critical in unit-oxidative damage [15,17,30]. Also, it was confirmed that C-phycocyanin binds with heavy metal ions [31]. The fluorescence quenching resulted in conformational changes in C-phycocyanin, possibly affecting the photosynthesis of algae due to charge capture mechanisms, as suggested by the study of Chi Z. et al. [32].

Carotenoids are integral components of the photosynthetic system, typically associated with one or more chlorophyll molecules through van der Waals forces. In addition to their crucial roles in light energy harvesting and dissipation of excess light energy, carotenoids also function as antioxidants and singlet oxygen quenchers, effectively reducing oxidative damage to the photosynthetic system and membrane lipids [21]. Our experimental results indicated that both the contents of carotenoid and chlorophyll a contents decreased with increasing As^3+^ concentrations (Figure 1c, d), suggesting that *A. platensis* initiated an antioxidant system in response to arsenic stress.

Additionally, the changes in the activity and content of SOD, CAT, and MDA reflected the antioxidant system’s response to As^3+^ stress: Under low arsenic stress, *A. platensis* cells activated the antioxidant defense mechanisms, resulting in significantly higher activities of SOD and CAT compared to the blank control group (Figure 3). This was consistent with the findings of Lyu, P et al. (2022) [33], who suggested that the enhancement of these enzyme activities is an adaptive response to oxidative stress damage. However, as the arsenic concentration further increased, the antioxidant system of *A. platensis* was damaged, i.e., structural disruption and reduced activity of SOD and CAT. The increase in MDA content indicated an exacerbation of membrane lipid peroxidation.

The transcriptomic analysis revealed a significant upregulation of the oxidative phosphorylation pathway (Figure 7a), with the expression levels of three NADH dehydrogenases (Hox family) also significantly increasing. The qPCR results further confirmed that the *A. platensis* cells enhanced the expression of Hox family genes (Figure 7b) to cope with the reactive oxygen species (ROS) generated under arsenic stress. Previous studies have shown that plants enhance stress tolerance by increasing endogenous hydrogen production and regulating antioxidant enzyme activity, thereby mitigating oxidative damage from heavy metal stress [34,35,36,37].

Additionally, the upregulation of Trxs gene expression in this study (Figure 9a) is consistent with previous findings [10]. Thioredoxins (Trxs) act as redox regulators in various biological processes, reducing thiol-containing peroxiredoxins (Prxs) and playing a critical role in ROS metabolism. The toxicity of As^3+^ primarily stems from its high thiol reactivity; As^3+^ has a strong affinity for -SH groups, inhibiting most -SH-containing enzymes and thereby causing harm to the organism. Therefore, the reduction in thiol-containing peroxidases is crucial [38]. Begum et al. [39] found that plants increase the activity of antioxidant enzymes to counteract the cellular damage caused by ROS induced by arsenic.

### 4.3. Mechanism of A. platensis’s Uptake and Transformation of As^3+^

Currently, the mechanisms of arsenic uptake and transformation in microorganisms and plants primarily include the oxidation of As^3+^, the reduction in As^5+^, the methylation of As^3+^, the demethylation of MAs^5+^, and arsenic efflux [40]. Relevant studies have shown that bacterial As^3+^ oxidases belong to the dimethyl sulfoxide (DMSO) reductase family, consisting of two subunits. The large subunit is a molybdenum-containing [3Fe-4S] cluster ferredoxin, while the small subunit is a [2Fe-2S] cluster ferredoxin [41]. Based on the transcriptome sequencing results (Figure 6b), the expression levels of ferredoxins were upregulated, indicating that one of the strategies of *A. platensis* cells to resist As^3+^ is to oxidize As^3+^. Both the methylation of As^3+^ and the demethylation of As^5+^ have been confirmed to be related to glutathione [38]. This study demonstrated that *A. platensis* cells increased the biosynthesis of glutathione under As^3+^ stress (Figure 6b), suggesting that *A. platensis* cells may possess detoxification mechanisms involving arsenic methylation and demethylation.

## 5. Conclusions

In this study, we investigated the response of *A. platensis* to As^3+^ stress through physiological and biochemical indicators as well as transcriptome sequencing analysis. We found that As^3+^ stress decreased levels of active substances in *A. platensis*, including phycocyanin and carotenoids. Additionally, the activity center of photosystem II was affected, hindering the electron transfer process. The activities of antioxidant enzymes, such as SOD and CAT, increased in response to the stress. Transcriptome analysis indicated that differentially expressed genes in *A. platensis* were primarily involved in processes such as ribosome and protein biosynthesis, regulation of cell wall organization and cell shape, and membrane transport, including transmembrane activities. From the transcriptome analysis, we identified that *A. platensis* resists As^3+^ stress through mechanisms such as arsenic redox reactions, arsenic methylation, and arsenic efflux. The qPCR results confirmed the expression levels of key stress-related genes in *A. platensis*, showing upregulation of Trx genes and the HoxEFU complex and downregulation of the cydB gene. This study provides valuable insights into the response mechanisms of *A. platensis* to arsenic stress and serves as a reference for addressing the issue of excessive arsenic in aquatic environments.

## Figures and Tables

**Figure 1 biology-13-01049-f001:**
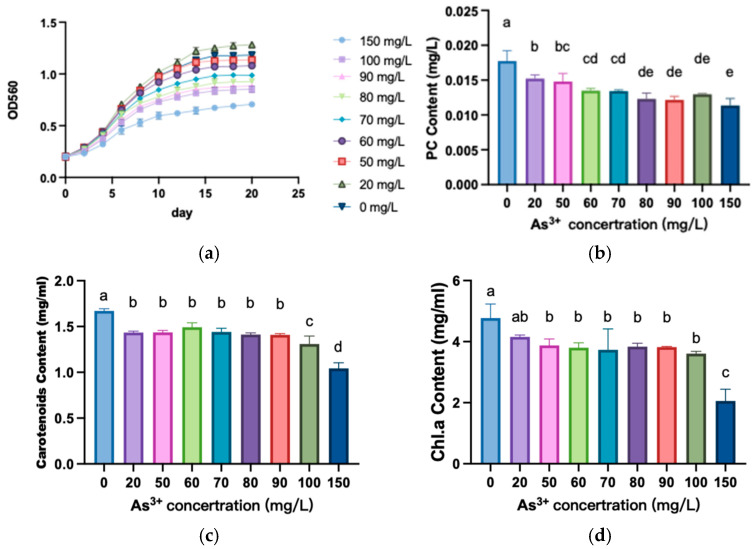
Effect of As^3+^ on the growth and biological activity concentration of *A. platensis*. (**a**) Growth of *A. platensis*. (**b**) PC content. (**c**) Carotenoids content. (**d**) Chlorophyll a content. Different letters indicate significant differences (*p* < 0.05).

**Figure 2 biology-13-01049-f002:**
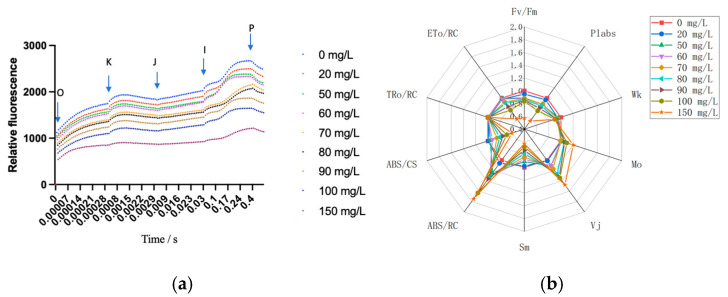
The photosynthetic system of *A. platensis*. (**a**) Results of OJIP fluorescence kinetics curves. (**b**) Chlorophyll fluorescence transients (Fv/Fm, ETo/RC, TRo/RC, ABS/CS, ABS/RC, Sm, Vj, Mo, Wk, and Plabs).

**Figure 3 biology-13-01049-f003:**
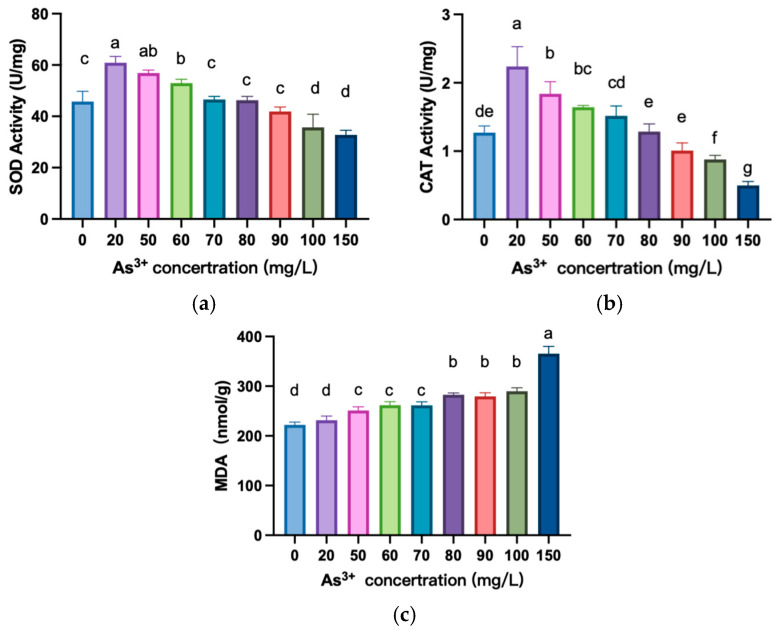
Effects of As^3+^ stress on the antioxidant system of *A. platensis*. (**a**) SOD activities. (**b**) CAT activities. (**c**) MDA content. Different letters indicate significant differences (*p* < 0.05).

**Figure 4 biology-13-01049-f004:**
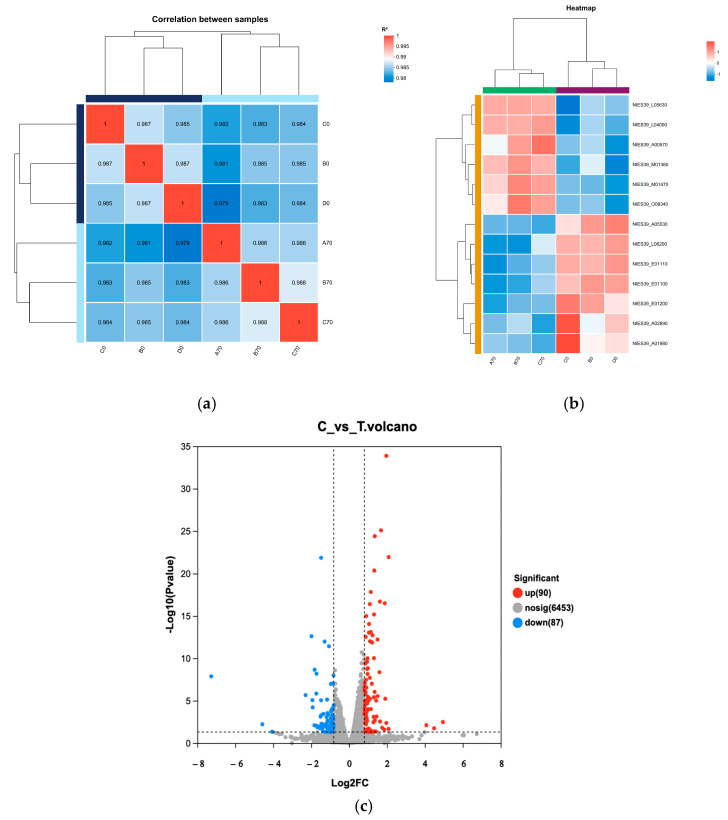
Transcriptome sequencing results. (**a**) Cluster analysis between samples. In the figure, the right and lower sides represent sample names, the left and upper sides are sample clustering, and the squares with different colors represent the correlation between the two samples (**b**) Hierarchically clustered heatmap of gene expression. Each column in the figure represents a sample, and each row represents a gene. The color in the heat map indicates the expression amount of the gene in the sample. See the number annotation under the color bar at the upper right for the specific expression amount change trend. (**c**) Volcano plot of differently expressed genes with *p* < 0.05 and |log2FC| ≥ 0.875.

**Figure 5 biology-13-01049-f005:**
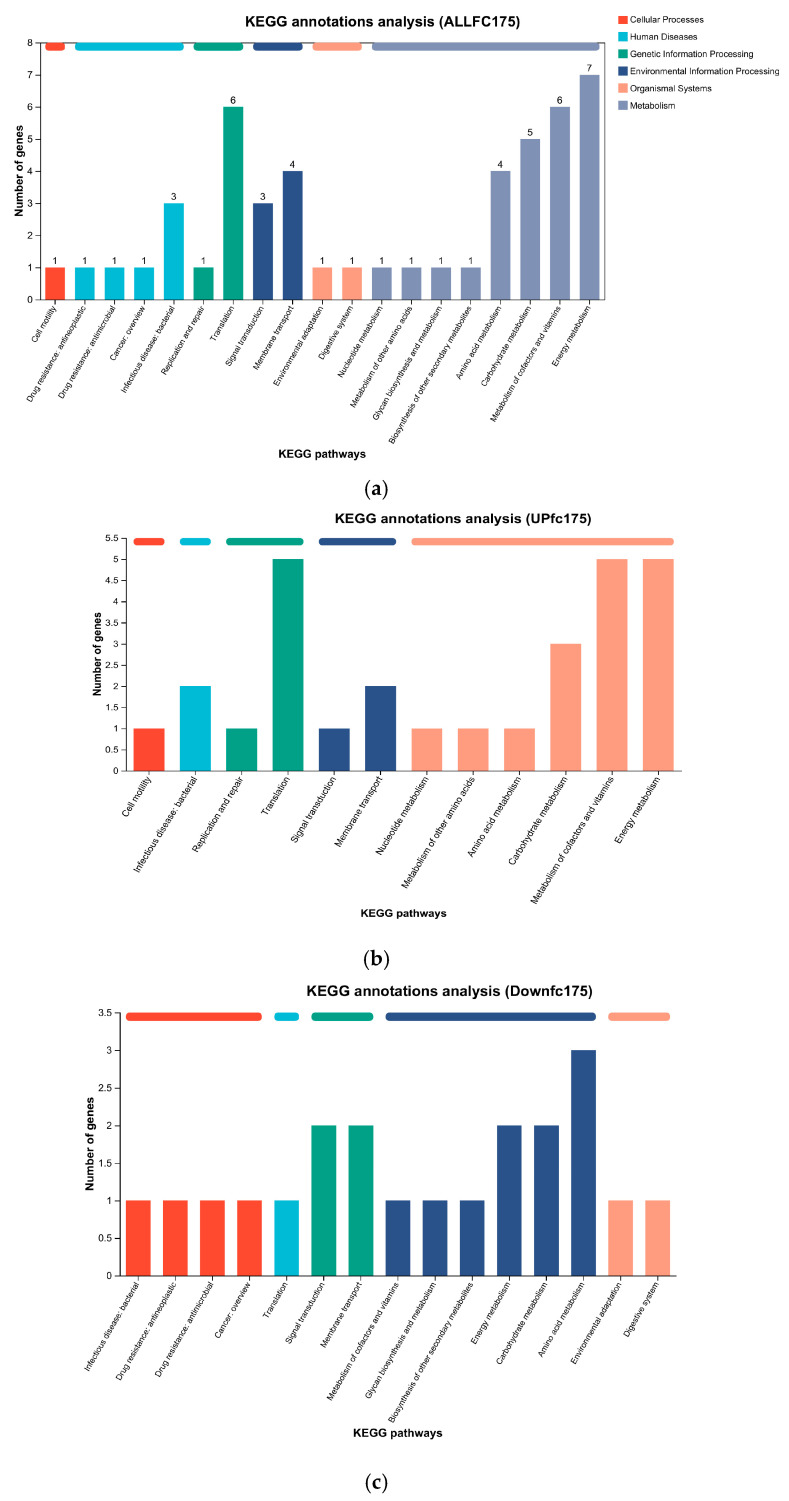
Results of KEGG enrichment analysis. The abscissa is the name of KEGG metabolic pathway, and the ordinate is the number of genes annotated to the pathway. See the legend for functional classification and grouping information. (**a**) KEGG enrichment analysis for all differentially expressed genes. (**b**) KEGG enrichment analysis for upregulated differentially expressed genes. (**c**) KEGG enrichment analysis for downregulated differentially expressed genes.

**Figure 6 biology-13-01049-f006:**
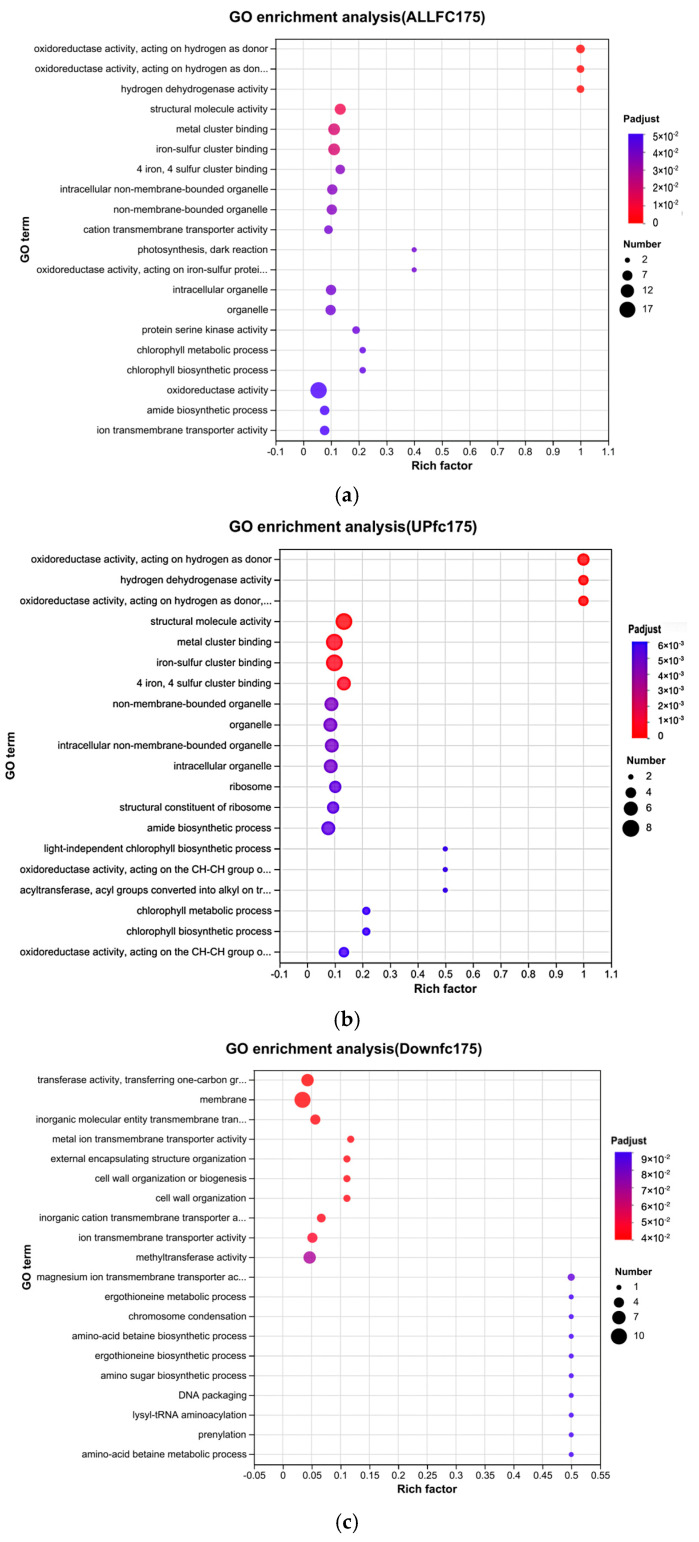
Results of GO functional enrichment analysis. The vertical axis represents GO term, and the horizontal axis represents rich factor [refers to the ratio of the number of genes annotated to the GO term in the gene set to the number of genes annotated to the GO term. The larger the rich factor, the greater the degree of enrichment]. The size of the point represents the number of genes/transcripts in this go term, and the color of the point corresponds to different FDR ranges. (**a**) GO enrichment analysis for all differentially expressed genes. (**b**) GO enrichment analysis for upregulated differentially expressed genes. (**c**) GO enrichment analysis for downregulated differentially expressed genes.

**Figure 7 biology-13-01049-f007:**
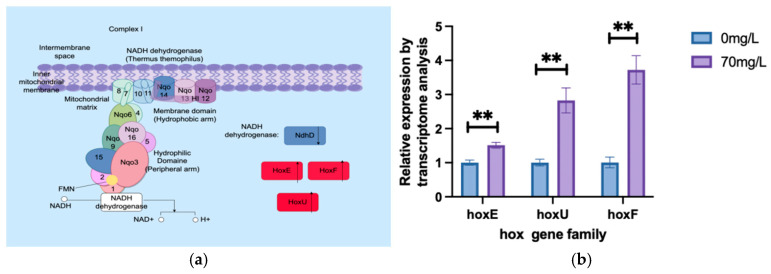
Oxidative phosphorylation pathway under As^3+^ stress. (**a**) Oxidative phosphorylation pathway. ( Red indicates upregulation, while blue indicates downregulation.) (**b**) qPCR results of the key genes of the oxidative phosphorylation. (** indicate significant differences (*p* < 0.05))

**Figure 8 biology-13-01049-f008:**
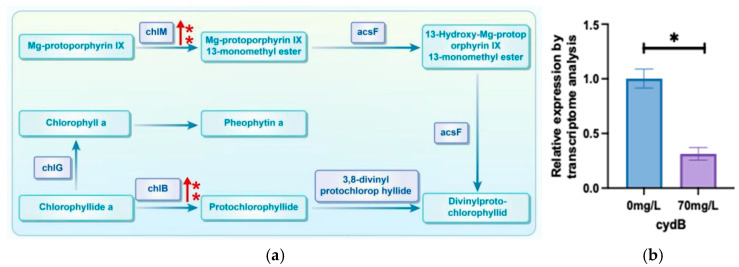
Phaeophytin a metabolism pathway under As^3+^ stress. (**a**) Phaeophytin a metabolism pathway. (the arrow indicates upregulation) (**b**) qPCR results of the key genes of the photosynthetic system. (* indicate significant differences (*p* < 0.05))

**Figure 9 biology-13-01049-f009:**
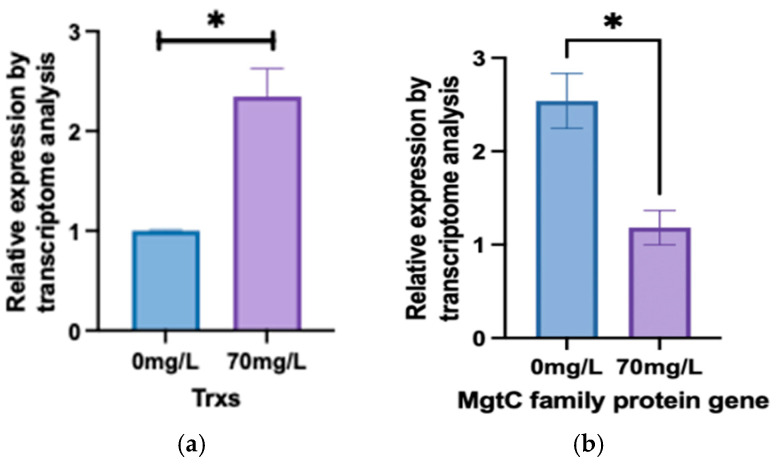
qPCR results. (**a**) qPCR result of the gene. (**b**) qPCR result of *MgtC family protein* gene. (* indicate significant differences (*p* < 0.05))

## Data Availability

All data generated or analyzed during this study are included in this published article.

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
