# Peer review of "Impact of Arsenic Stress on the Antioxidant System and Photosystem of Arthrospira platensis"

_biology, 2024, doi:10.3390/biology13121049_

Round 1
Reviewer 1 Report
Comments and Suggestions for Authors
The title is totally confusing.
Make it simple and clear. Basically you have analyzed redox state in
Arthrospira platensis against arsenic stress but representaing the work as the plant a s arsenic detoxicfier. So, mention a simpler title.
But I found that the abstract is very nicely described.
You need to work little more on the introduction
This part must describe three main things, 1. The current issues on arsenic and use of plants to mitigate from those issues, 2.. Work done on spirulina and arsenic issues, 3. Gap, and your hypothesis based on gap.
Materials methods
Ok
Results
OK
Discussion
Ok, but should not contain results
The main issue in the article is the gap. Arsenic induced changes in spirulina is studied long ago. So, you need to give a strong hypothesis.
All raw data need to be presented at the end of the article, may be as supplementary.
Author Response
|
Comments 1: The title is totally confusing. Make it simple and clear. Basically you have analyzed redox state in Arthrospira platensis against arsenic stress but representaing the work as the plant a s arsenic detoxicfier. So, mention a simpler title. |
||
|
Response 1: Thank you for pointing this out. We agree with this comment. Therefore, we. have replaced the title with “Impact of Arsenic Stress on the Antioxidant System and Photosystem of Arthrospira platensis” instead of “Anti oxidative stress injury mechanism as the main methods to response the stressed of arsenic in Arthrospira platensis”.(Page 1, Line 1)
|
Reviewer 2 Report
Comments and Suggestions for Authors
The manuscript focuses on an interesting topic concerning the mechanisms of response of A. platensis to arsenic stress.
Overall, the paper is quite carefully written, but I ask the authors to pay attention to several issues.
- In particular, some figures are only visible when the text is very “enlarged”: Figure 2b, Figure 4a and 4b, Figure 5, Figure 6;
- Please check the units (way of recording units of measurement) for all figures;
- please check figure captions - is the caption under figure 2 complete? Is the y-axis in Figure 2a captioned correctly? The caption for Figure 3 does not provide information on letters and significance (statistics) - of course this is explained below, but e.g. Figure 1 already has this information in the caption. Please standardise the captions;
- please check the units, in particular in lines: 76-78, 86-88, 90-91, 100, 103-105, 114-116, 156, 163-166, 185-187, 290, 321-322, 330
- The sentence on lines 271-272 is good for an introduction or discussion, not for the "Results" section;
- In line 311 it is suggested to write which specific species is meant, because such a notation suggests the whole genus Chlorella;
- In several places there are too many spaces or none at all.
Author Response
|
Comments 1: In particular, some figures are only visible when the text is very “enlarged”: Figure 2b, Figure 4a and 4b, Figure 5, Figure 6;.
|
|
Response 1: Thank you for pointing this out. We have completed the reprocessing of the figures. We adjusted the sharpness of the picture. |
|
Figure 2(b) Figure 4(a) Figure 4(b) Figure 5 Figure 6 Comments 2: Please check the units (way of recording units of measurement) for all figures;
|
|
Response 2: Agree. We have completed the inspection and correction of all figures units. We replaced”Fluorescence value” with “Relative fluorescence”(Figure 2(a)).(Page 5 line 271) We replaced “mg/mL” by “mg/ml”.(Page 5 line 255)
Comments 3: please check figure captions - is the caption under figure 2 complete? Is the y-axis in Figure 2a captioned correctly? The caption for Figure 3 does not provide information on letters and significance (statistics) - of course this is explained below, but e.g. Figure 1 already has this information in the caption. Please standardise the captions; Response 3: Thank you for pointing this out. We have revised the caption of Figure 2, corrected the y-axis label in Figure 2a (Page 5, Line 315), and added a legend in Figure 3 to indicate the significance of the letters. (Page 6 Line 345) Figure 2(a)
Comments 4: please check the units, in particular in lines: 76-78, 86-88, 90-91, 100, 103-105, 114-116, 156, 163-166, 185-187, 290, 321-322, 330 Response 4: We have changed all instances of "mL" to "ml." such as in lines:147,170-172,178,190-191,197.
Comments 5: The sentence on lines 271-272 is good for an introduction or discussion, not for the. "Results" section; |
Response 5: We have moved this sentence from the results section to the discussion section.We have. moved it from line 458-459 to line 590-592.
Comments 6: In line 311 it is suggested to write which specific species is meant, because such a notation。suggests the whole genus Chlorella;
Response 6: We have reviewed the literature and have changed “Chlorella” to “Anacystis nidulans” in line 472 (Page 12). Relevant studies have shown that short-term exposure to radionuclides led to a decrease in Fv/Fm and PIabs indices in Anacystis nidulans as the increase of exposure time and concentration [28], consistent with the results of this experiment.
Comments 7: In several places there are too many spaces or none at all.
Response 6: We have verified the formatting of all spaces. Such as line 473,480…

Reviewer 3 Report
Comments and Suggestions for Authors
The article “Anti-oxidative stress injury mechanism as the main method to respond to the stress of arsenic in Arthrospira platensis” is very relevant and interesting. The authors conducted a comprehensive study and presented the results well. The article can be printed in the journal after minor corrections.
General comments on the article:
1. Very many abbreviations and acronyms in the article which are not deciphered. The journal presents research of a wide profile in biology, so it is necessary to decipher the meanings of all terms to make it as clear as possible to a wide range of readers.
2. Correct the use of the name of the species Arthrospira platensis. The first time to use the full name and then the abbreviated A. platensis. In addition, the authors use the name Spirulina in the introduction and in Materials and Methods 2.2 and 2.3, and nowhere is there an explanation that A. platensis is meant. This should be corrected.
Other comments:
1. Title. The term “stress” is used twice in the title. This is not very good. Maybe replace the second time with the term “impact”.
2. Introduction. Lines 40-42. There is no literature reference for this statement.
3. Introduction. Lines 71. The introduction ends very abruptly. There is information about what is already known about the issue, while there is no information and what is not yet known. Why the authors started to research this particular aspect. And then the purpose of this research should be added.
3- Materials of the study. P.2.1 “Strains” should be replaced by “Strain”. Only one strain is studied.
4. Line 77. In the form of which compound was arsenic added to the growth medium.
5. Line 79. “OD=0.2.” Add which day of growth.
6. Line 80-82. The phrase is not very clear. Correct it.
7. Line 80. Why is the term Spirulina suddenly used here and below.
8. Line 84 and 279. Why is the year mentioned in parentheses?
9. Line 83. Section 2.3 The section describes the definition of phycocyanin only, not “active ingredients”.
10. Line 111, 117. Decipher PS II, SOD, CAT, MDA
11. Line 151. Figure 1 (c) the abbreviation MS should be decoded. Figure 1c - The abbreviation “Car” looks bad.
12. Lines 170-179. Decipher the parameters that are given as abbreviations.
13. Lines 180. Decipher “SOP” and “CAT”.
14. Figure 4: The legend is made in a very small font. It should be enlarged. Figure caption is in different font. Caption Figure 4, 5, 6 to give more explanatory information.
15. Figure 5 b, c and figure 6 b, c - make larger. The font in the figures is not clearly visible.
16. Lines 271-274. It is not clear where this is seen in figure 8. Is it more likely to be figure 10?
17. You could combine Figures 9 and 10 into one figure.
Comments on the Quality of English LanguageA minor correction to the English language is required. Some phrases are not very clear.
Author Response
|
Comments 1: Very many abbreviations and acronyms in the article which are not deciphered. The journal presents research of a wide profile in biology, so it is necessary to decipher the meanings of all terms to make it as clear as possible to a wide range of readers. |
|
Response 1: Thank you for pointing this out. We have deciphered the abbreviations such as, we deciphered “PS II” to “photosystem II”in line 151(Page 3) and in line 157-158(Page 3),we deciphered “SOD” to “Superoxide dismutase “,deciphered “CAT” to “Catalase”,deciphered “MDA” to” Malondialdehyde”. |
|
Comments 2: Correct the use of the name of the species Arthrospira platensis. The first time to use the full name and then the abbreviated A. platensis. In addition, the authors use the name Spirulina in the introduction and in Materials and Methods 2.2 and 2.3, and nowhere is there an explanation that A. platensis is meant. This should be corrected. |
|
Response 2: Agree. We have corrected the use of the name of the species Arthrospira platensis. Such as Page1 line 25,28,33, Page 2 line 68,69,71,74 ,Page 3 line 123,135,152, Page 4 line 254,Page 5 line 256,257,259,260,265,271,272,277,283, Page 6 line 298,300,306,Page 10 line 360,363,365,372-373,377,383,392,Page 11 line 402,406,409,420,422,Page 12 line 445,447,456,467,476,477,480,487,489,Page 13 line 529,532,538,543,558,566,568,569,572, Page 14 line 603,607,610, 612,614.
Comments 3: Title. The term “stress” is used twice in the title. This is not very good. Maybe replace the second time with the term “impact”. Response 3: Thank you for pointing this out. We agree with this comment. Therefore, we. have replaced the title with “Impact of Arsenic Stress on the Antioxidant System and Photosystem of Arthrospira platensis” instead of “Anti oxidative stress injury mechanism as the main methods to response the stressed of arsenic in Arthrospira platensis”. (Page 1 line1)
Comments 4: Introduction. Lines 40-42. There is no literature reference for this statement. Response 4: We have added literature reference (reference 1)for the statement in Page 1 line 40-42.
Comments 5: Introduction. Lines 71. The introduction ends very abruptly. There is information about what is already known about the issue, while there is no information and what is not yet known. Why the authors started to research this particular aspect. And then the purpose of this research should be added. |
Response 5: We have added the fourth paragraph identifies the gaps in current research and. articulates our research objectives and main findings in introduction. (Page2 line88-99)
Comments 6: Materials of the study. P.2.1 “Strains” should be replaced by “Strain”. Only one strain is studied.
Response 6: We have replaced “Strains” by “Strain” in line 101(Page 2).
Comments 7: Line 77. In the form of which compound was arsenic added to the growth medium.
Response 7: We have detailed the specific arsenic components added to the culture medium in. line 116. (Page 3) For stress treatments, sodium arsenite standard solution at initial concentrations ranging from 0 to 150 mg/L was added to the medium when the cells reached OD560 = 0.2 ± 0.01(Day 1).
Comments 8: Line 79. “OD=0.2.” Add which day of growth.
Response 8: We have added which day of growth in line 117. (Page 3) For stress treatments, sodium arsenite standard solution at initial concentrations ranging from 0 to 150 mg/L was added to the medium when the cells reached OD560 = 0.2 ± 0.01(Day 1).
Comments 9: Line 80-82. The phrase is not very clear. Correct it.
Response 9: We have corrected it in line 120-122(Page 3). The UV spectrophotometer was used. to measure OD560 every two days for 20 days (0 Day, 2Day, 4Day, 6Day…20Day) and plot the growth curve with days as the X-axis and OD560 value as the Y-axis.
Comments 10: Line 80. Why is the term Spirulina suddenly used here and below.
Response 10: We have replaced “Spirulina” by “A. platensis”. 2.2. Determination of A. platensis. growth curve. (Page3 line119.)
Comments 11: Line 84 and 279. Why is the year mentioned in parentheses?
Response 11: We have removed the year. The determination of phycocyanin was conducted according to Han [5].(Page 3 line 124)This finding is consistent with the results of Gan et al. [22].(Page 11 line 422-423)
Comments 12: Line 83. Section 2.3 The section describes the definition of phycocyanin only, not。“active ingredients”.
Response 12: We have adjusted the arrangement in our methodology and added the composition. of photosynthetic pigments. (Page 3 line 123-144)
2.3. Determination of active substances in A. platensis cells
The determination of phycocyanin was conducted according to Han [5]. Specifically, Algae cultures in the logarithmic growth phase (Day 8) were adjusted to a uniform OD, and 15 ml aliquots were taken. These aliquots were then centrifuged (12000 r/min, 4 ℃, 15 min) to collect the algal pellet in 50 ml centrifuge tubes. Subsequently, 15 ml of PBS buffer (0.1 mol/L, pH 7.8) was added to the pellet, which was then vortexed. The mixture was subjected to repeated freeze-thaw cycles at -80 ℃ and 25 ℃, with fixed durations (30 min at -80 ℃, 15 min at 25 ℃), repeated 6 times. After the final cycle, the sample was centrifuged again (12000 r/min, 4 ℃, 10 min), and the supernatant was collected for measurement of OD615 and OD652 to calculate the concentration of phycocyanin.
PC(mg/ml)=0.187×OD615-0.089×OD652 [14]
Extraction of Chlorophyll a and Carotenoids from A. platensis using Organic Solvent Immersion Method [5]. The algal pellets collected were the same as 2.3 in 50 ml centrifuge tubes. 20 ml of 96% ethanol was added, and the mixture was vigorously shaken. The tubes were then wrapped in foil and placed in a 4 ℃ refrigerator for 24 hours, with intermittent shaking every 6 hours. The samples were centrifuged again (8000 r/min, 4 ℃, 10 min), and the supernatants' OD665, OD649, and OD470 were measured to calculate the concentrations of chlorophyll a and carotenoids using the formula provided in reference [15].
Chlorophyll a:Ca (mg/L) =9.78×4OD662-0.990×OD644
Chlorophyll b:Cb (mg/L) =21.426×OD644-4.650×OD663
Carotenoids:Cc (mg/L) =4.695×OD440-0.268×(Ca+Cb)
Comments 13: Line 111, 117. Decipher PS II, SOD, CAT, MDA
Response 13: We have deciphered PS II, SOD, CAT and MDA in line 151,157-158. We deciphered “PS II” to “photosystem II”in line 151(Page 3) and in line 157-158(Page 3), we deciphered “SOD” to “Superoxide dismutase”,deciphered “CAT” to “Catalase”,deciphered “MDA” to” Malondialdehyde”.
Comments 14: Line 151. Figure 1 (c) the abbreviation MS should be decoded. Figure 1c - The. abbreviation “Car” looks bad.
Response 14: We have adjusted and corrected the figure 1c accordingly. We replaced "car" with
“Carotenoids”(Page 5 line 255)
Comments 15: Lines 170-179. Decipher the parameters that are given as abbreviations.
Response 15: We have categorized and elaborated on these abbreviations of proprietary terms in. line 278-279. Chlorophyll fluorescence transients (Fv/Fm, ETo/RC, TRo/RC, ABS/CS, ABS/RC, Sm, Vj, Mo, Wk and Plabs.) ( Page 5 line 278-179)
Comments 16: Lines 180. Decipher “SOP” and “CAT”
Response 16: We have deciphered “SOD” and “CAT” in Page 3 line 157-158. The suspension was used. to measure the activities of Superoxide dismutase (SOD) and Catalase (CAT), as well as the intracellular Malondialdehyde (MDA)
Comments 17: Figure 4: The legend is made in a very small font. It should be enlarged. Figure. caption is in different font. Caption Figure 4, 5, 6 to give more explanatory information.
Response 17: We have adjusted the figure 4 and gave figure 4, 5, 6 more explanatory information.(Page 7 line. 315-322) (Page 8 line 331-33) (Page 9-10, line 350-354)
- (b)
(c)
Figure 4. Transcriptome sequencing results. (a) Cluster analysis between samples. In the figure, the right and lower sides represent sample names, the left and upper sides are sample clustering, and the squares with different colors represent the correlation between the two samples (b) Hierarchically clustered heatmap of gene expression. Each column in the figure represents a sample, and each row represents a gene. The color in the heat map indicates the expression amount of the gene in the sample. See the number annotation under the color bar at the upper right for the specific expression amount change trend. (c) Volcano plot of differently expressed genes with p < 0.05 and |log2FC| ≥ 0.875. (Page 7 line. 315-322)
Figure 5. Results of KEGG enrichment analysis. The abscissa is the name of KEGG metabolic pathway, and the ordinate is the number of genes annotated to the pathway. See the legend for functional classification and grouping information. (a) KEGG enrichment analysis for all differentially expressed genes. (b) KEGG enrichment analysis for up-regulated differentially expressed genes. (c) KEGG enrichment analysis for down-regulated differentially expressed genes. (Page 8 line 331-33)
Figure 6. Results of GO functional enrichment analysis. The vertical axis represents GO term, and the horizontal axis represents rich factor [refers to the ratio of the number of genes annotated to the GO term in the gene set to the number of genes annotated to the GO term. The larger the rich factor, the greater the degree of enrichment]. The size of the point represents the number of genes / transcripts in this go term, and the color of the point corresponds to different FDR ranges. (a) GO enrichment analysis for all differentially expressed genes. (b) GO enrichment analysis for up-regulated differentially expressed genes. (c) GO enrichment analysis for down-regulated differentially expressed genes. (Page 9-10, line 350-354)
Comments 18: Figure 5 b, c and figure 6 b, c - make larger. The font in the figures is not clearly. visible.
Response 18: We have adjusted these figures accordingly.
Figure 5 (b)
Figure 5 (c)
Figure 6 (b)
Figure 6 (c)
Comments 19: Lines 271-274. It is not clear where this is seen in figure 8. Is it more likely to be figure 10?
Response 19: Yes, we have corrected it.We combine Figures 9 and 10 into Figure 9.(Page 11 line411).
Comments 20: You could combine Figures 9 and 10 into one figure.
Response 20: Thank you for pointing this out. We agree with this comment. Therefore, we have. combined them into figure 9.
Figure 9

Round 2
Reviewer 1 Report
Comments and Suggestions for Authors
The ms is much improved, but any self-citation must be fixed. You can cite 2-3 of your own work.
Comments on the Quality of English LanguageLanguage is ok.